# GRAPPA: GRAMMAR-AUGMENTED PRE-TRAINING FOR TABLE SEMANTIC PARSING

**Tao Yu**[†][*]**, Chien-Sheng Wu**[§]**, Xi Victoria Lin**[§][*]**, Bailin Wang**[‡][*]**, Yi Chern Tan**[†]**, Xinyi Yang**[§]**,**
**Dragomir Radev**[†][§]**, Richard Socher**[§]**, Caiming Xiong**[§]

[§] Salesforce Research, [†] Yale University, [‡]University of Edinburgh

{tao.yu, yichern.tan,dragomir.radev}@yale.edu, bailin.wang@ed.ac.uk
{wu.jason,x.yang,rsocher,cxiong}@salesforce.com, victorialin@fb.com

## ABSTRACT

We present GRAPPA, an effective pre-training approach for table semantic parsing that learns a compositional inductive bias in the joint representations of textual and tabular data. We construct synthetic question-SQL pairs over high-quality tables via a synchronous context-free grammar (SCFG). We pre-train GRAPPA on the synthetic data to inject important structural properties commonly found in table semantic parsing into the pre-training language model. To maintain the model's ability to represent real-world data, we also include masked language modeling (MLM) on several existing table-and-language datasets to regularize our pre-training process. Our proposed pre-training strategy is much data-efficient. When incorporated with strong base semantic parsers, GRAPPA achieves new state-of-the-art results on four popular fully supervised and weakly supervised table semantic parsing tasks. The pre-trained embeddings can be downloaded at https://huggingface.co/Salesforce/grappa_large_jnt.

## 1 INTRODUCTION

Tabular data serve as important information source for human decision makers in many domains, such as finance, health care, retail and so on. While tabular data can be efficiently accessed via the structured query language (SQL), a natural language interface allows such data to be more accessible for a wider range of non-technical users. As a result, table semantic parsing that maps natural language queries over tabular data to formal programs has drawn significant attention in recent years.

Recent pre-trained language models (LMs) such as BERT (Devlin et al., 2019) and RoBERTa (Liu et al., 2019) achieve tremendous success on a spectrum of natural language processing tasks, including semantic parsing (Zettlemoyer & Collins, 2005; Zhong et al., 2017; Yu et al., 2018b). These advances have shifted the focus from building domain-specific semantic parsers (Zettlemoyer & Collins, 2005; Artzi & Zettlemoyer, 2013; Berant & Liang, 2014; Li & Jagadish, 2014) to cross-domain semantic parsing (Zhong et al., 2017; Yu et al., 2018b; Herzig & Berant, 2018; Dong & Lapata, 2018; Wang et al., 2020; Lin et al., 2020).

Despite such significant gains, the overall performance on complex benchmarks such as SPIDER (Yu et al., 2018b) and WIKITABLEQUESTIONS benchmarks are still limited, even when integrating representations of current pre-trained language models. As such tasks requires generalization to new databases/tables and more complex programs (e.g., SQL), we hypothesize that current pre-trained language models are not sufficient for such tasks. First, language models pre-trained using unstructured text data such as Wikipedia and Book Corpus are exposed to a significant domain shift when directly applied to table semantic parsing, where jointly modeling the relation between utterances and structural tables is crucial. Second, conventional pre-training objectives does not consider the underlying compositionality of data (e.g., questions and SQLs) from table semantic parsing. To close this gap, we seek to learn contextual representations jointly from structured tabular data and unstructured natural language sentences, with objectives oriented towards table semantic parsing.

---

[*] This work was mostly done during Tao and Bailin's internship at Salesforce Research. Victoria is now at Facebook AI.

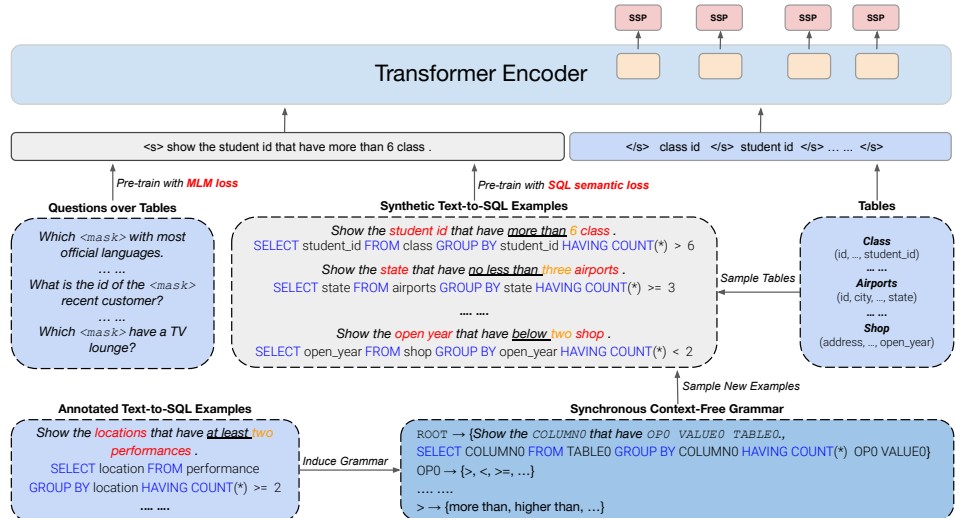

Figure 1: An overview of GRAPPA pre-training approach. We first induce a SCFG given some examples in SPIDER. We then sample from this grammar given a large amount of tables to generate new synthetic examples. Finally, GRAPPA is pre-trained on the synthetic data using SQL semantic loss and a small amount of table related utterances using MLM loss.

In this paper, we propose a novel grammar-augmented pre-training framework for table semantic parsing (GRAPPA). Inspired by previous work on data synthesis for semantic parsing (Berant & Liang, 2014; Wang et al., 2015b; Jia & Liang, 2016; Herzig & Berant, 2018; Andreas, 2020), we induce a synchronous context-free grammar (SCFG) specific to mapping natural language to SQL queries from existing text-to-SQL datasets, which covers most commonly used question-SQL patterns. As shown in Figure 1, from a text-to-SQL example we can create a question-SQL template by abstracting over mentions of schema components (tables and fields), values, and SQL operations. By executing this template on randomly selected tables we can create a large number of synthetic question-SQL pairs. We train GRAPPA on these synthetic question-SQL pairs and their corresponding tables using a novel text-schema linking objective that predicts the syntactic role of a table column in the SQL for each pair. This way we encourage the model to identify table schema components that can be grounded to logical form constituents, which is critical for most table semantic parsing tasks.

To prevent overfitting to the synthetic data, we include the masked-language modelling (MLM) loss on several large-scale, high-quality table-and-language datasets and carefully balance between preserving the original natural language representations and injecting the compositional inductive bias through our synthetic data. We pre-train GRAPPA using 475k synthetic examples and 391.5k examples from existing table-and-language datasets. Our approach dramatically reduces the training time and GPU cost. We evaluate on four popular semantic parsing benchmarks in both fully supervised and weakly supervised settings. GRAPPA consistently achieves new state-of-the-art results on all of them, significantly outperforming all previously reported results.

## 2 METHODOLOGY

### 2.1 MOTIVATION

Semantic parsing data is compositional because utterances are usually related to some formal representations such as logic forms and SQL queries. Numerous prior works (Berant & Liang, 2014; Wang et al., 2015a; Jia & Liang, 2016; Iyer et al., 2017; Andreas, 2020) have demonstrated the benefits of augmenting data using context-free grammar. The augmented examples can be used to teach the model to generalize beyond the given training examples.

However, data augmentation becomes more complex and less beneficial if we want to apply it to generate data for a random domain. More and more work (Zhang et al., 2019b; Herzig et al., 2020b; Campagna et al., 2020; Zhong et al., 2020) shows utilizing augmented data doesn't always result in a

significant performance gain in cross-domain semantic parsing end tasks. The most likely reason for this is that models tend to overfit to the canonical input distribution especially the generated utterances are very different compared with the original ones.

Moreover, instead of directly training semantic parsers on the augmented data, our paper is the first to use the synthetic examples in pre-training in order to inject a compositional inductive bias to LMs and show it actually works if the overfitting problem is carefully addressed. To address the overfitting problem, in Section 2.3, we also include a small set of table related utterances in our pre-training data. We add an MLM loss on them as a regularization factor, which requires the model to balance between real and synthetic examples during the pre-training. We note that this consistently improves the performance on all downstream semantic parsing tasks (see Section 4). Finally, our pre-training method is much more data-efficient and save much more computational power than other prior work (Section 5).

## 2.2 DATA SYNTHESIS WITH SYNCHRONOUS CONTEXT-FREE GRAMMAR

We follow Jia & Liang (2016) to design our SCFG and apply it on a large amount of tables to populate new examples. For example, as shown in Figure 1, by replacing substitutable column mentions ("locations"), table mentions ("performance"), values ("two"), and SQL logic phrases ("at least") with the other possible candidates in the same group, our grammar generates new synthetic text-to-SQL examples with the same underlying SQL logic template. We then pre-train BERT on the augmented examples to force it to discover substitutable fragments and learn the underlying logic template so that it is able to generalize to other similar questions. Meanwhile, BERT also benefits from pre-training on a large number of different columns, table names, and values in the generated data, which could potentially improve schema linking in semantic parsing tasks.

| Non-terminals | Production rules |
|---|---|
| TABLE $\rightarrow t_i$
COLUMN $\rightarrow c_i$
VALUE $\rightarrow v_i$
AGG $\rightarrow \langle$ MAX, MIN, COUNT, AVG, SUM $\rangle$
OP $\rightarrow \langle =, \leq, \neq, ... ,$ LIKE, BETWEEN $\rangle$
SC $\rightarrow \langle$ ASC, DESC $\rangle$
MAX $\rightarrow \langle$ "maximum", "the largest"... $\rangle$
$\leq \rightarrow \langle$ "no more than", "no above"... $\rangle$
... | 1. ROOT $\rightarrow \langle$ "For each COLUMN0 , return how many times TABLE0 with COLUMN1 OP0 VALUE0 ?",
`SELECT COLUMN0 , COUNT ( * ) WHERE COLUMN1 OP0 VALUE0 GROUP BY COLUMN0` $\rangle$

2. ROOT $\rightarrow \langle$ "What are the COLUMN0 and COLUMN1 of the TABLE0 whose COLUMN2 is OP0 AGG0 COLUMN2 ?",
`SELECT COLUMN0 , COLUMN1 WHERE COLUMN2 OP0 ( SELECT AGG0 ( COLUMN2 ) )` $\rangle$ |

Table 1: Examples of non-terminals and production rules in our SCFG. Each production rule ROOT $\rightarrow \langle \alpha, \beta \rangle$ is built from some $(x, y) \in \mathcal{D}$ by replacing all terminal phrases with non-terminals. $t_i$, $c_i$, and $v_i$ stand for any table name, column name, entry value respectively.

**Grammar induction** To induce a cross-domain SCFG, we study examples in SPIDER since it is a publicly available dataset that includes the largest number of examples with complex compositionalities in different domains. To further show the generality of our approach, we do not develop different SCFG for each downstream task. Given a set of $(x, y)$ pairs in SPIDER, where $x$ and $y$ are the utterance and SQL query respectively. We first define a set of non-terminal symbols for table names, column names, cell values, operations, etc. For example, in Table 1, we group aggregation operations such as MAX as a non-terminal AGG. We can also replace the entities/phrases with their non-terminal types in SQL query to generate a SQL production rule $\beta$. Then, we group $(x, y)$ pairs by similar SQL production rule $\beta$. We automatically group and count Spider training examples by program templates, and select about 90 most frequent program templates $\beta$. For each program template in the grammar, we randomly select roughly 4 corresponding natural language questions, manually replace entities/phrases with their corresponding non-terminal types to create natural language templates $\alpha$, and finally align them to generate each production rule ROOT $\rightarrow \langle \alpha, \beta \rangle$. The manual alignment approximately takes a few hours. About 500 SPIDER examples are studied to induce the SCFG.

**Data augmentation** With $\langle \alpha, \beta \rangle$ pairs, we can simultaneously generate pseudo natural questions and corresponding SQL queries given a new table or database. We first sample a production rule, and replace its non-terminals with one of corresponding terminals. For example, we can map the non-terminal AGG to MAX and "maximum" for the SQL query and the natural language sentence, respectively. Also, table content is used in synthesizing our pre-training data. For example, if the sampled production rule contains a value (e.g., VALUE0), we sample a value for the selected column

from the table content and add it to the SQL and question templates. This way during pre-training, GRAPPA can access the table content and learn the linking between values and columns.

We use WIKITABLES (Bhagavatula et al., 2015), which contains 1.6 million high-quality relational Wikipedia tables. We remove tables with exactly the same column names and get about 340k tables and generate 413k question-SQL pairs given these tables. Also, we generate another 62k question-SQL pairs using tables and databases in the training sets of SPIDER and WIKISQL. In total, our final pre-training dataset includes 475k question-SQL examples.

We note that SCFG is usually crude (Andreas, 2020) especially when it is applied to augment data for different domains. In this work we don't focus on how to develop a better SCFG that generates more natural utterances. We see this as a very interesting future work to explore. Despite the fact that the SCFG is crude, our downstream task experiments show that it could be quite effective if some pre-training strategies are applied.

## 2.3 TABLE RELATED UTTERANCES

As discussed in Section 2.1, GRAPPA is also pre-trained on human annotated questions over tables with a MLM objective. We collected seven high quality datasets for textual-tabular data understanding (Table 8 in the Appendix), all of them contain Wikipedia tables or databases and the corresponding natural language utterances written by humans. We only use tables and contexts as a pre-training resource and discard all the other human labels such as answers and SQL queries.

## 2.4 PRE-TRAINING GRAPPA

Unlike all the previous work where augmented data is used in the end task training, we apply the framework to language model pre-training. Training semantic parsers is usually slow, and augmenting a large amount of syntactic pairs directly to the end task training data can be prohibitively slow or expensive. In our work, we formulate text-to-SQL as a multi-class classification task for each column, which can be naturally combined with the MLM objective to pre-train BERT for semantic parsing. Moreover, in this way, the learned knowledge can be easily and efficiently transferred to downstream semantic parsing tasks in the exact same way as BERT (shown in Section 4).

GRAPPA is initialized by RoBERTa$_{\text{LARGE}}$ (Liu et al., 2019) and further pre-trained on the synthetic data with SQL semantic loss and table-related data with MLM loss. As shown in Figure 1, we follow Hwang et al. (2019) to concatenate a user utterance and the column headers into a single flat sequence separated by the `` token. The user utterance can be either one of the original human utterances collected from the aggregated datasets or the canonical sentences sampled from the SCFG. We add the table name at the beginning of each column if there are some complex schema inputs involving multiple tables. We employ two objective functions for language model pre-training: 1) masked-language modelling (MLM), and 2) SQL semantic prediction (SSP).

**MLM objective**   Intuitively, we would like to have a self-attention mechanism between natural language and table headers. We conduct masking for both natural language sentence and table headers. A small part of the input sequence is first replaced with the special token `<mask>`. The MLM loss is then computed by the cross-entropy function on predicting the masked tokens. We follow the default hyperparameters from Devlin et al. (2019) with a 15% masking probability.

**SSP objective**   With our synthetic natural language sentence and SQL query pairs, we can add an auxiliary task to train our column representations. The proposed task is, given a natural language sentence and table headers, to predict whether a column appears in the SQL query and what operation is triggered. We then convert all SQL sequence labels into operation classification labels for each column. For example in the Figure 1, the operation classification label of the column "locations" is SELECT AND GROUP BY HAVING. In total, there are 254 potential classes for operations in our experiments.

For a column or table indexed by $i$, we use the encoding of the special token `` right before it as its representation, denoted as $\mathbf{x_i}$ to predict its corresponding operations. On top of such representations, we apply a two-layer feed-forward network followed by a GELU activation layer (Hendrycks & Gimpel, 2016) and a normalization layer (Ba et al., 2016) to the output representations. Formally, we

compute the final vector representation of each column $\mathbf{y_i}$ by:

$$\mathbf{h} = \text{LayerNorm}(\text{GELU}(W_1 \cdot \mathbf{x_i}))$$
$$\mathbf{y_i} = \text{LayerNorm}(\text{GELU}(W_2 \cdot \mathbf{h}))$$

Finally, $\mathbf{y_i}$ is employed to compute the cross-entropy loss through a classification layer. We sum losses from all columns in each training example for back-propagation. For samples from the aggregated datasets, we only compute the MLM loss to update our model. For samples from the synthetic data we generated, we compute only SSP loss to update our model. More specifically, we mix 391k natural language utterances and 475k synthetic examples together as the final pre-training data. The examples in these two groups are randomly sampled during the pre-training, and MLM loss is computed if the selected example is a natural language question, otherwise SSP for a synthetic example.

## 3 EXPERIMENTS

We conduct experiments on four *cross-domain* table semantic parsing tasks, where generalizing to unseen tables/databases at test time is required. We experiment with two different settings of table semantic parsing, fully supervised and weakly supervised setting. The data statistics and examples on each task are shown in Table 2 and Table 7 in the Appendix respectively.

| Task & Dataset | # Examples | Resource | Annotation | Cross-domain |
|---|---|---|---|---|
| SPIDER Yu et al. (2018b) | 10,181 | database | SQL | ✓ |
| Fully-sup. WIKISQL Zhong et al. (2017) | 80,654 | single table | SQL | ✓ |
| WIKITABLEQUESTIONS Pasupat & Liang (2015) | 2,2033 | single table | answer | ✓ |
| Weakly-sup. WIKISQL Zhong et al. (2017) | 80,654 | single table | answer | ✓ |

Table 2: Overview of four table-based semantic parsing and question answering datasets in fully-supervised (top) and weakly-supervised (bottom) setting used in this paper. More details in Section 3

### 3.1 SUPERVISED SEMANTIC PARSING

We first evaluate GRAPPA on two supervised semantic parsing tasks. In a supervised semantic parsing scenario, given a question and a table or database schema, a model is expected to generate the corresponding program.

**SPIDER** SPIDER Yu et al. (2018b) is a large text-to-SQL dataset. It consists of 10k complex question-query pairs where many of the SQL queries contain multiple SQL keywords. It also includes 200 databases where multiple tables are joined via foreign keys. For the baseline model, we use RAT-SQL + BERT Wang et al. (2020) which is the state-of-the-art model according to the official leaderboard. We followed the official Spider evaluation to report set match accuracy.

**Fully-sup. WIKISQL** WIKISQL Zhong et al. (2017) is a collection of over 80k questions and SQL query pairs over 30k Wikipedia tables. We use Guo & Gao (2019), a competitive model on WIKISQL built on SQLova Hwang et al. (2019), as our base model. We adapt the same set of hyperparameters including batch size and maximum input length as in Guo & Gao (2019). For a fair comparison, we only consider single models without execution-guided decoding and report execution accuracy.

### 3.2 WEAKLY-SUPERVISED SEMANTIC PARSING

We also consider weakly-supervised semantic parsing tasks, which are very different from SQL-guided learning in pre-training. In this setting, a question and its corresponding answer are given, but the underlying meaning representation (e.g., SQL queries) are unknown.

**WIKITABLEQUESTIONS** This dataset contains question-denotation pairs over single Wikipedia tables Pasupat & Liang (2015). The questions involve a variety of operations such as comparisons, superlatives, and aggregations, where some of them are hard to answered by SQL queries.

| Models | Dev. | Test |
|---|---|---|
| Global-GNN (Bogin et al., 2019) | 52.7 | 47.4 |
| EditSQL (Zhang et al., 2019b) | 57.6 | 53.4 |
| IRNet (Guo et al., 2019) | 61.9 | 54.7 |
| RYANSQL (Choi et al., 2020) | 70.6 | 60.6 |
| TranX (Yin et al., 2020a) | 64.5 | - |
| RAT-SQL (Wang et al., 2019) | 62.7 | 57.2 |
| w. BERT-large | 69.7 | 65.6 |
| w. RoBERTa-large | 69.6 | - |
| w. GRAPPA (MLM) | 71.1(+1.4) | - |
| w. GRAPPA (SSP) | **73.6(+3.9)** | 67.7(+2.1) |
| w. GRAPPA (MLM+SSP) | **73.4(+3.7)** | **69.6(+4.0)** |

Table 3: Performance on SPIDER. We run each model three times by varying random seeds, and the average scores are shown.

| Models | Dev. | Test |
|---|---|---|
| (Dong & Lapata, 2018) | 79.0 | 78.5 |
| (Shi et al., 2018) | 84.0 | 83.7 |
| (Hwang et al., 2019) | 87.2 | 86.2 |
| (He et al., 2019) | 89.5 | 88.7 |
| (Lyu et al., 2020) | 89.1 | 89.2 |
| (Guo & Gao, 2019) | 90.3 | 89.2 |
| w. RoBERTa-large | 91.2 | 90.6 |
| w. GRAPPA (MLM) | 91.4 | 90.7 |
| w. GRAPPA (SSP) | 91.2 | 90.7 |
| w. GRAPPA (MLM+SSP) | 91.2 | **90.8** |
| w. RoBERTa-large (10k) | 79.6 | 79.2 |
| w. GRAPPA (MLM+SSP) (10k) | **82.3(+2.7)** | **82.2(+3.0)** |

Table 4: Performance on fully-sup. WIKISQL. All results are on execution accuracy without execution-guided decoding.

We used the model proposed by Wang et al. (2019) which is the state-of-the-art parser on this task. This model is a two-stage approach that first predicts a partial "abstract program" and then refines that program while modeling structured alignments with differential dynamic programming. The original model uses GloVe Pennington et al. (2014) as word embeddings. We modified their implementation to encode question and column names in the same way as we do in our fine-tuning method that uses RoBERTa and GRAPPA.

**Weakly-sup. WIKISQL**   In the weakly-supervised setting of WIKISQL, only the answers (i.e., execution results of SQL queries) are available. We also employed the model proposed by Wang et al. (2019) as our baseline for this task. We made the same changes and use the same experiment settings as described in the previous section for WIKITABLEQUESTIONS.

### 3.3   IMPLEMENTATION OF GRAPPA

For fine-tuning RoBERTa, we modify the code of RoBERTa implemented by Wolf et al. (2019) and follow the hyperparameters for fine-tuning RoBERTa on RACE tasks and use batch size 24, learning rate $1e$-5, and the Adam optimizer Kingma & Ba (2014). We fine-tune GRAPPA for 300k steps on eight 16GB Nvidia V100 GPUs. The pre-training procedure can be done in less than 10 hours. For all downstream experiments using GRAPPA or RoBERTa, we always use a BERT specific optimizer to fine-tune them with a learning rate of $1e$-5, while using a model-specific optimizer with the respective learning rate for the rest of the base models.

## 4   EXPERIMENTAL RESULTS

We conducted experiments to answer the following two questions: 1) Can GRAPPA provide better representations for table semantic parsing tasks? 2) What is the benefit of two pre-training objectives, namely MLM and SSP? Since GRAPPA is initialized by RoBERTa, we answer the first question by directly comparing the performance of base parser augmented with GRAPPA and RoBERTa on table semantic parsing tasks. For the second question, we report the performance of GRAPPA trained with MLM, SSP and also a variant with both of them (MLM+SSP).

**Overall results**   We report results on the four aforementioned tasks in Tables 3, 4, 5, and 6 respectively. Overall, base models augmented with GRAPPA significantly outperforms the ones with RoBERTa by 3.7% on SPIDER, 1.8% on WIKITABLEQUESTIONS, and 2.4% on weakly-sup. WIKISQL, and achieve new state-of-the-art results across all four tasks. In most cases, the combined objective of MLM+SSP helps GRAPPA achieve better performance when compared with independently using MLM and SSP. Moreover, on the low-resource setting, GRAPPA outperforms RoBERTa by 3.0% in fully-sup. WIKISQL and 3.9% in WIKITABLEQUESTIONS. Detailed results for each task are discussed as follows.

**SPIDER**   Results on SPIDER are shown in Table 3. When augmented with GRAPPA, the model achieves significantly better performance compared with the baselines using BERT and RoBERTa.

| Models | Dev. | Test |
|---|---|---|
| (Liang et al., 2018) | 42.3 | 43.1 |
| (Dasigi et al., 2019) | 42.1 | 43.9 |
| (Agarwal et al., 2019) | 43.2 | 44.1 |
| (Herzig et al., 2020b) | - | 48.8 |
| (Yin et al., 2020b) | 52.2 | 51.8 |
| (Wang et al., 2019) | 43.7 | 44.5 |
| w. RoBERTa-large | 50.7(+7.0) | 50.9(+6.4) |
| w. GRAPPA (MLM) | 51.5(+7.8) | 51.7(+7.2) |
| w. GRAPPA (SSP) | 51.2(+7.5) | 51.1(+6.6) |
| w. GRAPPA (MLM+SSP) | **51.9(+8.2)** | **52.7(+8.2)** |
| w. RoBERTa-large ×10% | 37.3 | 38.1 |
| w. GRAPPA (MLM+SSP) ×10% | **40.4(+3.1)** | **42.0(+3.9)** |

Table 5: Performance on WIKITABLEQUESTIONS. Results trained on 10% of the data are shown at the bottom.

| Models | Dev. | Test |
|---|---|---|
| (Liang et al., 2018) | 72.2 | 72.1 |
| (Agarwal et al., 2019) | 74.9 | 74.8 |
| (Min et al., 2019) | 84.4 | 83.9 |
| (Herzig et al., 2020b) | 85.1 | 83.6 |
| (Wang et al., 2019) | 79.4 | 79.3 |
| w. RoBERTa-large | 82.3 (+2.9) | 82.3 (+3.0) |
| w. GRAPPA (MLM) | 83.3 (+3.9) | 83.5 (+4.2) |
| w. GRAPPA (SSP) | 83.5(+4.1) | 83.7 (+4.4) |
| w. GRAPPA (MLM+SSP) | **85.9 (+6.5)** | **84.7 (+5.4)** |

Table 6: Performance on weakly-sup. WIKISQL. We use (Wang et al., 2019) as our base model.

Our best model, GRAPPA with MLM+SSP achieves the new state-of-the-art performance, surpassing previous one (RAT-SQL+BERT-large) by a margin of 4%. Notably, most previous top systems use pre-trained contextual representations (e.g., BERT, TaBERT), indicating the importance of such representations for the cross-domain parsing task.

**Fully sup. WIKISQL**   Results on WIKISQL are shown in Table 4. All GRAPPA models achieve nearly the same performance as RoBERTa. We suspect it is the relatively large training size and easy SQL pattern of WIKISQL make the improvement hard, comparing to SPIDER. Hence, we set up a low-resource setting where we only use 10k examples from the training data. As shown in the bottom two lines of Table 4, GRAPPA improves the performance of the SQLova model by 3.0% compared to RoBERTa, indicating that GRAPPA can make the base parser more sample-efficient.

**WIKITABLEQUESTIONS**   Results on WIKITABLEQUESTIONS are shown in Table 5. By using RoBERTa and GRAPPA to encode question and column inputs, the performance of Wang et al. (2019) can be boosted significantly ( >6%). Compared with RoBERTa, our best model with GRAPPA (MLM+SSP) can further improve the performance by 1.8%, leading to a new state-of-the-art performance on this task. Similar to the low-resource experiments for WIKISQL, we also show the performance of the model when trained with only 10% of the training data. As shown at the bottom two lines Table 5, GRAPPA (MLM + SSP) obtains much better performance than RoBERTa, again showing its superiority of providing better representations.

**Weakly sup. WIKISQL**   Results on weakly supervised WIKISQL are shown in Table 6. GRAPPA with MLM+SSP again achieves the best performance when compared with other baselines, obtain the new state-of-the-art results of 84.7% on this task. It is worth noting that our best model here is also better than many models trained in the fully-supervised setting in Table 4. This suggests that inductive biases injected in pre-trained representation of GRAPPA can significantly help combat the issue of spurious programs introduced by learning from denotations Pasupat & Liang (2015); Wang et al. (2019) when gold programs are not available.

## 5   ANALYSIS

**Pre-training objectives**   GRAPPA trained with both MLM and SSP loss consistently outperforms the one trained with one of them (MLM+SSP vs. MLM only or SSP only). GRAPPA (MLM) usually improves the performance by around 1% such as 1.4% gain on SPIDER (dev), 0.8% on WIKITABLEQUESTIONS, and 1.2% on weakly-sup. WIKISQL. By pre-training on the synthetic text-to-SQL examples, GRAPPA (SSP), we can see a similar performance gain on these tasks too except 3.9% improvement on SPIDER dev, which is what we expected (grammar is overfitted to SPIDER). By pre-training with both MLM and SSP on the combined data, GRAPPA (MLM+SSP) consistently and significantly outperforms the one pre-trained with MLM or SSP separately (e.g., about +2% on Spider, +1.5% on WikiTableQuestions, and +1.2% on weakly-sup WikiSQL.). This contributes to our key argument in the paper: in order to effectively inject compositional inductive bias to LM, pre-training on synthetic data should be regularized properly (using SSP+MLM together

instead of SSP or MLM only) in order to balance between preserving the original BERT encoding ability and injecting compositional inductive bias, otherwise, the improvements are not robust and limited (using SSP or MLM only).

**Generalization** As mentioned in Section 2.2, we design our SCFG solely based on SPIDER, and then sample from it to generate synthetic examples. Despite the fact that GRAPPA pre-trained on such corpus is optimized to the SPIDER data distribution, which is very different from WIKISQL and WIKITABLEQUESTIONS, GRAPPA is still able to improve performance on the two datasets. In particular, for WIKITABLEQUESTIONS where the underlying distribution of programs (not necessarily in the form of SQL) are latent, GRAPPA can still help a parser generalize better, indicating GRAPPA can be beneficial for general table understanding even though it is pre-trained on SQL specific semantics. We believe that incorporating rules from a broader range of datasets (e.g. WIKITABLEQUESTIONS) would further improve the performance. However, in this paper, we study rules from only the SPIDER dataset and test the effectiveness on other unseen datasets with different different underlying rules on purpose in order to show the generality of our method.

Even though GRAPPA is pre-trained on synthetic text-to-SQL data, the proposed pre-training method can also be applied to many other semantic parsing tasks with different formal programs (e.g., logic forms); and we also demonstrated the effectness of GRAPPA on non text-to-SQL tasks (weakly-supervised WIKISQL and WIKITABLEQUESTIONS where no programs are used, training is supervised by only answers/cell values) the underlying distribution of programs (not necessarily in the form of SQL) are latent. Furthermore, to design the SCFG and synthesize data with the corresponding programs labeled, we can use any formal programs such as the logic form or SParQL, and then employ the data to pre-train GraPPa. In this paper we choose SQL as the formal program to represent the formal representation of the questions simply because more semantic parsing datasets are labeled in SQL.

**Pre-training time and data** Our experiments on the SPIDER and WIKITABLEQUESTIONS tasks show that longer pre-training doesn't improve and can even hurt the performance of the pre-trained model. This also indicates that synthetic data should be carefully used in order to balance between preserving the original BERT encoding ability and injecting compositional inductive bias. The best result on SPIDER is achieved by using GRAPPA pre-trained for only 5 epochs on our relatively small pre-training dataset. Compared to other recent pre-training methods for semantic parsing such as TaBERT (Yin et al., 2020a) and TAPAS (Herzig et al., 2020a), GRAPPA achieves the state-of-the-art performance (incorporated with strong base systems) on the four representative table semantic parsing tasks *in less 10 hours on only 8 16GB Nvidia V100 GPUs* (6 days on more than 100 V100 GPUs and 3 days on 32 TPUs for TaBERT and TAPAS respectively) Moreover, we encourage future work on studying how the size and quality of synthetic data would affect the end task performance. Also, GRAPPA (MLM+SSP) consistently outperforms other settings, which indicates that using MLM on the human annotated data is important.

**Pre-training vs. training data augmentation** Many recent work (Zhang et al., 2019b; Herzig et al., 2020b; Campagna et al., 2020; Zhong et al., 2020) in semantic parsing and dialog state tracking show that training models on a combination of the extra synthetic data and original training data does not improve or even hurt the performance. For example, (Zhong et al., 2020) synthesize data on training databases in several semantic parsing tasks including SPIDER, and find that training with this data augmentation leads to overfitting on the synthetic data and decreases the performance. In contrast, our pre-training approach could effectively utilize a large amount of synthesized data and improve downstream task performance. Also, the base parser with a GRAPPA encoder could usually converge to a higher performance in shorter time (see Section A.1).

## 6 RELATED WORK

**Textual-tabular data understanding** Real-world data exist in both structured and unstructured forms. Recently the field has witnessed a surge of interest in joint textual-tabular data understanding problems, such as table semantic parsing (Zhong et al., 2017; Yu et al., 2018b), question answering (Pasupat & Liang, 2015; Chen et al., 2020), retrieval (Zhang et al., 2019a), fact-checking (Chen et al., 2019) and summarization (Parikh et al., 2020; Radev et al., 2020). While most work focus on

single tables, often obtained from the Web, some have extended modeling to more complex structures such as relational databases (Finegan-Dollak et al., 2018; Yu et al., 2018b; Wang et al., 2020). All of these tasks can benefit from better representation of the input text and different components of the table, and most importantly, an effective contextualization across the two modalities. Our work aims at obtaining high-quality cross-modal representation via pre-training to potentially benefit all downstream tasks.

**Pre-training for NLP tasks**  GRAPPA is inspired by recent advances in pre-training for text such as (Devlin et al., 2019; Liu et al., 2019; Lewis et al., 2020b;a; Guu et al., 2020). Seminal work in this area shows that textual representation trained using conditional language modeling objectives significantly improves performance on various downstream NLP tasks. This triggered an exciting line of research work under the themes of (1) cross-modal pre-training that involves text (Lu et al., 2019; Peters et al., 2019; Yin et al., 2020a; Herzig et al., 2020a) and (2) pre-training architectures and objectives catering subsets of NLP tasks (Lewis et al., 2020b;a; Guu et al., 2020). GRAPPA extends these two directions further. The closest work to ours are TaBERT (Yin et al., 2020a) and TAPAS (Herzig et al., 2020a). Both are trained over millions of web tables and relevant but noisy textual context. In comparison, GRAPPA is pre-trained with a novel training objective, over synthetic data plus a much smaller but cleaner collection of text-table datasets.

**Data augmentation for semantic parsing**  Our work was inspired by existing work on data augmentation for semantic parsing (Berant & Liang, 2014; Wang et al., 2015a; Jia & Liang, 2016; Iyer et al., 2017; Yu et al., 2018a). Berant & Liang (2014) employed a rule-based approach to generate canonical natural language utterances given a logical form. A paraphrasing model was then used to choose the canonical utterance that best paraphrases the input and to output the corresponding logical form. In contrast, Jia & Liang (2016) used prior knowledge in structural regularities to induce an SCFG and then directly use the grammar to generate more training data, which resulted in a significant improvement on the tasks. Unlike these works which augment a relatively small number of data and use them directly in end task training, we synthesize a large number of texts with SQL logic grounding to each table cheaply and use them for pre-training.

## 7 CONCLUSION AND FUTURE WORK

In this paper, we proposed a novel and effective pre-training approach for table semantic parsing. We developed a context-free grammar to automatically generate a large amount of question-SQL pairs. Then, we introduced GRAPPA, which is an LM that is pre-trained on the synthetic examples with SQL semantic loss. We discovered that, in order to better leverage augmented data, it is important to add MLM loss on a small amount of table related utterances. Results on four semantic parsing tasks demonstrated that GRAPPA significantly outperforms RoBERTa.

While the pre-training method is surprisingly effective in its current form, we view these results primarily as an invitation for more future work in this direction. For example, this work relies on a hand-crafted grammar which often generates unnatural questions; Further improvements are likely to be made by applying more sophisticated data augmentation techniques. Also, it would be interesting to study the relative impact of the two objectives (MLM and SSP) by varying the respective number of pre-training examples. Furthermore, pre-training might benefit from synthesizing data from a more compositional grammar with a larger logical form coverage, and also from supervising by a more compositional semantic signals.

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

# A  APPENDICES

| Task | Question | Table/Database | Annotation |
|---|---|---|---|
| SPIDER | Find the first and last names of the students who are living in the dorms that have a TV Lounge as an amenity. | database with 5 tables e.g. `student`, `dorm_amenity`, ... | SELECT T1.FNAME, T1.LNAME FROM STUDENT AS T1 JOIN LIVES_IN AS T2 ON T1.STUID=T2.STUID WHERE T2.DORMID IN ( SELECT T3.DORMID FROM HAS_AMENITY AS T3 JOIN DORM_AMENITY AS T4 ON T3.AMENID=T4.AMENID WHERE T4.AMENITY_NAME= 'TV LOUNGE') |
| Fully-sup. WIKISQL | How many CFL teams are from York College? | a table with 5 columns e.g. `player`, `position`, ... | SELECT COUNT CFL TEAM FROM CFLDRAFT WHERE COLLEGE = 'YORK' |
| WIKITABLEQUESTIONS | In what city did Piotr's last 1st place finish occur? | a table with 6 columns e.g. `year`, `event`, ... | "Bangkok, Thailand" |
| Weakly-sup. WIKISQL | How many CFL teams are from York College? | a table with 5 columns e.g. `player`, `position`,... | 2 |

Table 7: Examples of the inputs and annotations for four semantic parsing tasks. SPIDER and Fully-sup. WIKISQL require full annotation of SQL programs, whereas WIKITABLEQUESTIONS and Weakly-sup. WIKISQL only requires annotation of answers (or denotations) of questions.

| | Train Size | # Table | Task |
|---|---|---|---|
| TabFact | 92.2K | 16K | Table-based fact verification |
| LogicNLG | 28.5K | 7.3K | Table-to-text generation |
| HybridQA | 63.2K | 13K | Multi-hop question answering |
| WikiSQL | 61.3K | 24K | Text-to-SQL generation |
| WikiTableQuestions | 17.6K | 2.1K | Question answering |
| ToTTo | 120K | 83K | Table-to-text generation |
| Spider | 8.7K | 1K | Text-to-SQL generation |

Table 8: Aggregated datasets for table-and-language tasks.

## A.1  ADDITIONAL ANALYSIS

**Training coverage**  As shown in Figure 2, on the challenging end text-to-SQL SPIDER task, RAT-SQL initialized with GRAPPA outperforms RAT-SQL using RoBERTa by about 14% in the early training stage. This shows that GRAPPA already captures some semantic knowledge in pre-training. Finally, GRAPPA is able to keep the competitive edge by 4%.

**What if the task-specific training data is also used with the MLM or SSP objective in pre-training?**  Although we did not do the same experiments, we would like to point to the RAT-SQL paper (Wang et al., 2020) for some suggestions. They add a similar alignment loss (similar to SSP) on the SPIDER training data and found that it doesn't make a statistically significant difference (in Appendix B).

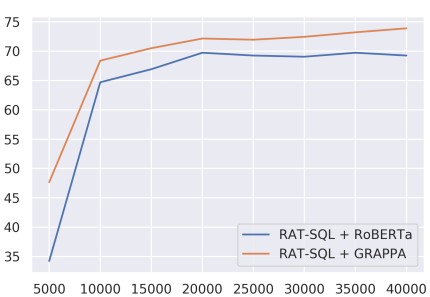

Figure 2: The development exact set match score in SPIDER vs. the number of training steps. RAT-SQL initialized with our pre-trained GRAPPA converges to higher scores in a shorter time than RAT-SQL $w$. BERT.

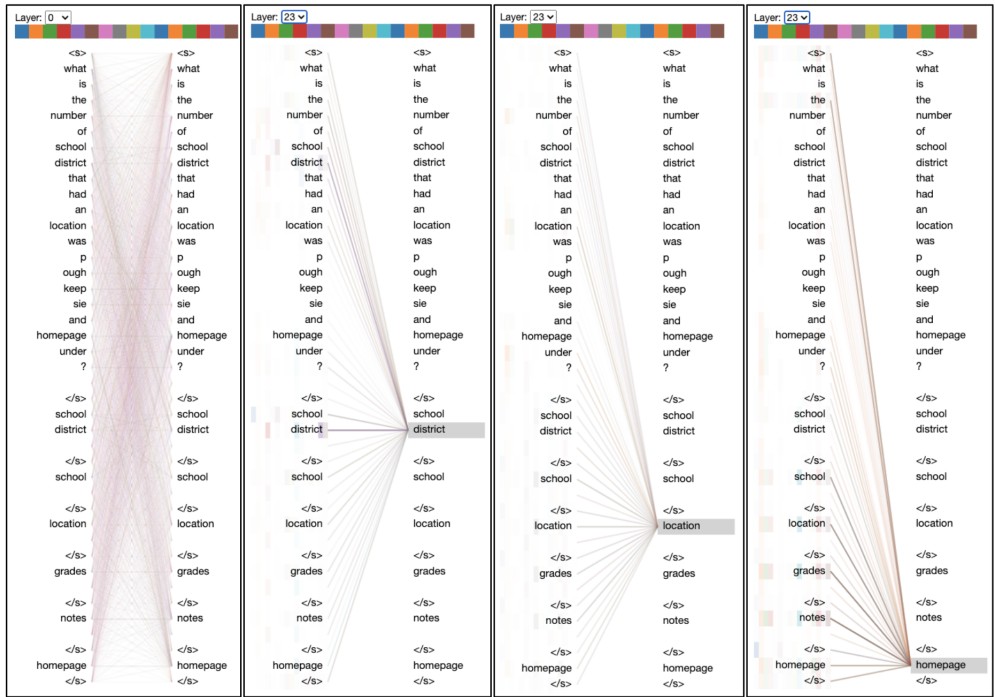

(a) RoBERTa-large

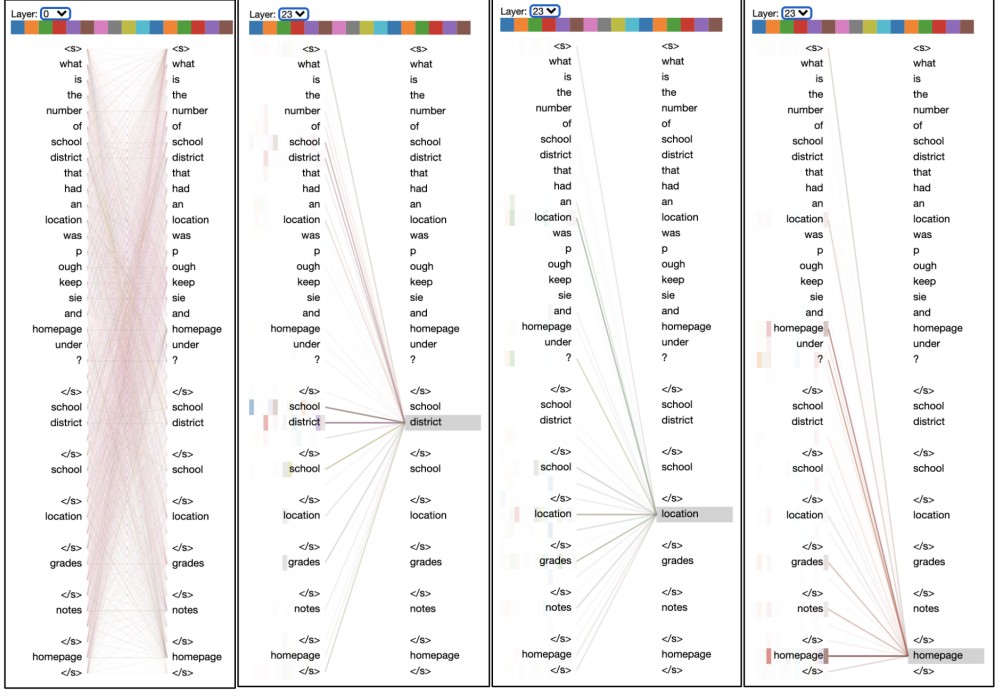

(b) GRAPPA

Figure 3: Attention visualization on the last self-attention layer.

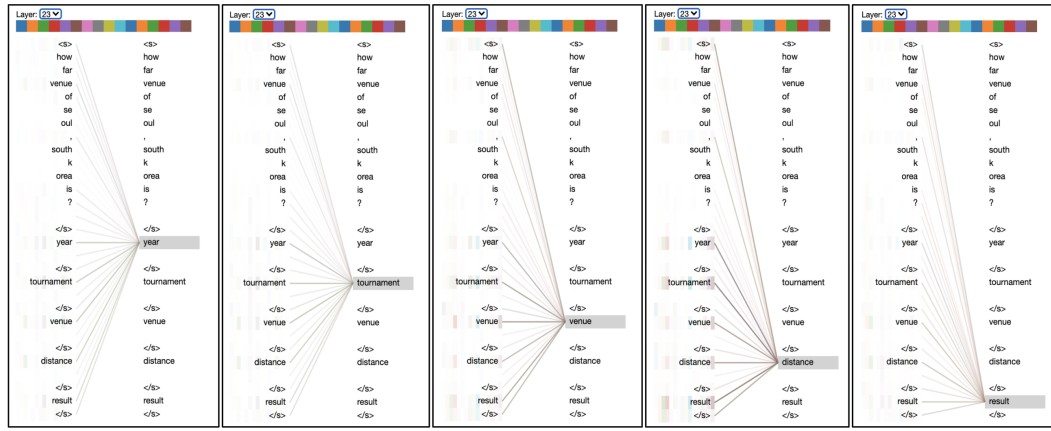

(a) RoBERTa-large

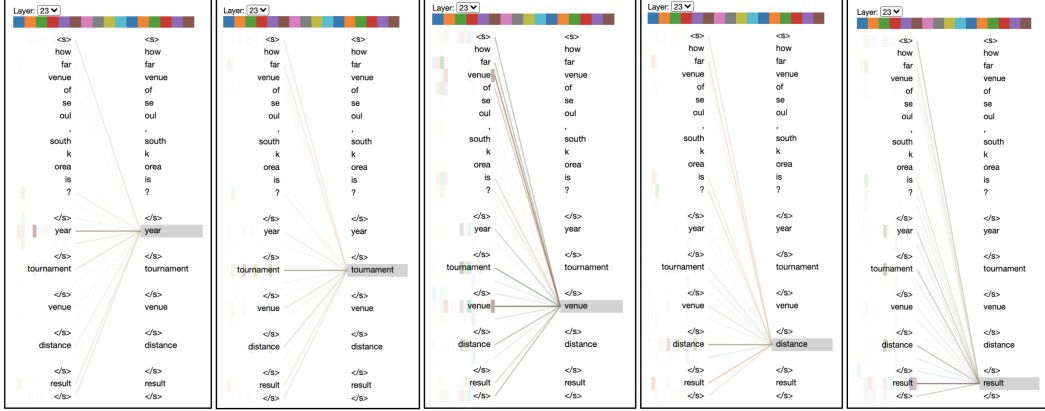

(b) GRAPPA

Figure 4: Attention visualization on the last self-attention layer.

