# OpenReview forum: "GraPPa: Grammar-Augmented Pre-Training for Table Semantic Parsing"
_ICLR.cc/2021/Conference — ICLR 2021 Poster_

### Official Review · AnonReviewer4 · 2020-10-28
**Not too novel, but a good solid result**

**Rating:** 7
**Confidence:** 4

**Review:**

This work explores a pretraining strategy (similar to https://arxiv.org/abs/1606.03622) to the problem of table question answering. More specifically a synchronous context-free grammar (SCFG) is first learned from training data (with manual alignment of entities/phrases). Then the SCFG is used to generate more full supervision data for Roberta model pretraining. The training objective is a combination of two parts: SQL Semantic Precision (SSP) predicts elements in SQL given the question on the synthetic data, and masked-language modeling (MLM) on the natural (training) data.

Experiment on 3 dataset shows significant improvement over RoBERTa baseline (especially when using only 10% of training data). New state-of-the art results are achieved. Oblation study shows the impact of  SSP, MLM, training time, dataset etc., which is nice.

---

> ### Author Response · Authors · 2020-11-16
> **Thank you for your review**
>
> Thank you for your feedback and a nice summary of our work!
>
> We would like to further elaborate on the novelty of our paper. The key problem in semantic parsing is the lack of data. There is a lot of work on semantic parsing ((Berant and Liang, 2014), (Jia and Liang, 2016), and (Andreas, 2020), and so on) trying to synthesize more data using different methods. However, more and more work shows that models do NOT always benefit from data augmentation especially when the original data is relatively large and cross-domain (Zhang et al., 2019b; Herzig et al., 2020b; Campagna et al., 2020; Zhong et al., 2020). One possible reason is that, under a large scale training setting, direct training with the augmented data tends to overfit on the synthetic data.
>
> Instead of directly training semantic parsers on the augmented data, our paper is the *first to use it in pre-training in order to inject a compositional inductive bias to LMs and show it actually works if a critical pre-training trick* (pre-trained with MLM+SSP instead of MLM or SSP only, pre-train for only a few number of epochs to avoid overfitting to the synthetic data) is applied. This pre-training method is *much more data- and GPU-efficient than other work* (e.g. only <10 hours on 8 V100 GPUs vs. 6 days on >100 V100 GPUs). Please find more details in our response to Reviewer 3. Thanks!

---

### Official Review · AnonReviewer3 · 2020-10-28
**lack of novelty, augmented data is homologous with benchmarks used for testing, improvements are marginal**

**Rating:** 5
**Confidence:** 4

**Review:**

##########################################################################

Summary:


The paper provides a interesting direction in pre-training for table semantic parsing. In particular, it proposes to first collect a collection of pseudo question-SQL pairs in an automatic way, based on tables from WikiTables tables and tables and databases in the training sets  SPIDER and WIKISQL. After that, masked language modeling and a newly introduced task called SSP, which is, given a natural language sentence and table headers, to predict whether a column appears in the SQL query and what operation is triggered. Experiments on Spider and WikiSQL show that the model achieves new state-of-the-art.

##########################################################################

Reasons for score:


Overall, I vote for rejection. I like the idea of pre-training for table semantic parsing.  My major concerns are the novelty of the method and the experiment.
1. The merit of a pre-trained model is its ability to horizontally support (to some extend) related tasks, not only limited by one. However, the pre-training conducted in this work is different from standard unsupervised pre-training in that the data used for pre-training is task-specific, i.e. for the task of natural language to SQL generation, and the evaluation is only conducted on natural language to SQL generation tasks. I agree that NL-to-SQL generation is a semantic parsing task, but semantic parsing is a broad area (see Percy Liang's CACM survey), only limited to NL-to-SQL generation.
2. Even though the authors pre-train on table-based NL-to-SQL generation, as a pre-trained model, the model should be tested on other semantic parsing problems. Taking one step back, even though tasks are limited to NL-to-SQL generation, at least experiment should be conducted on a dataset whose tables are not used to produce pseudo dataset which is used in the pre-training stage.
3. It is worth to note that the pseudo data used for pre-training comes from the training data of the benchmark datasets used for evaluation. I agree that with the data augmentation process, the pre-trained model can see many newly composed questions and SQLs. However, the development of such a pre-trained seems to be too tailored for the task and for the datasets. In such a tailored way, the improvements are also marginal to me. From 4 tables, except for Table 3, SSP performs comparable to MLM, which reflects that the importance of the standard masked language modeling.
4. The novelty of the methodology is limited. There is no improvements in terms of neural model architecture. The pre-trained task seems trivial and too specific for SQL generation.

---

> ### Author Response · Authors · 2020-11-16
> **Thank you for your review**
>
> Thanks for your feedback! Please find as follows our response to your questions below.
>
> **Novelty of the paper**
> We would like to emphasize that although our work at first glance appears to be a straightforward usage of synthetic data for semantic parsing tasks, showing the points below is *non-trivial*:
> 1. *Finding a unique pre-training strategy*: It is the *first* paper showing pre-training on a combination of natural text and synthesized data with MLM+SSP objectives could significantly improve the SOTA performance on four popular and representative (full- and weakly-sup.) semantic parsing tasks (E.g., +3.7% on Spider, +1.8% on WikiTableQuestions, and +2.4% on weakly-sup. WikiSQL in Table 3-6).
> 2. *Identifying key pre-training details to maximally benefit performance*: People might think pre-training on synthetic data could simply improve performance on semantic parsing tasks. We show the improvement is limited if this is done naively (typically +<1% from our SSP-only models). *In order to effectively inject compositional inductive bias to LM, pre-training on synthetic data should balance between preserving the language modeling ability and injecting compositional inductive bias*. More specifically, GraPPa pre-trained with MLM+SSP consistently performs much better than the one pre-trained with MLM or SSP separately (MLM+SSP vs. MLM only or SSP only: E.g., about +2% on Spider, +1.5% on WikiTableQuestions, and +1.2% on weakly-sup WikiSQL.). Our experiments on the Spider task also show that longer pre-training doesn’t improve and can even hurt the performance of the pre-trained model. In our experiments we pre-train for only ~5 epochs, otherwise, it will hurt the performance. We performed many pre-training and downstream task experiments variations to identify such important details.
> 3. *Our pre-training strategy for semantic parsing is much more time- and energy-efficient than prior work*: GraPPa achieves SOTA on the four tasks by pre-training for *only <10 hours on 8 V100 GPUs compared with 6 days on >100 V100 GPUs for TaBERT and 3 days on 32 TPUs for TAPAS.* This could inspire a lot of future work in this direction without prohibitive requirements on computational power.
>
> **Is our method specifically designed for and only tested on text-to-SQL tasks?**
> As Reviewer 1 indicated, our pre-training method can also be applied to many other semantic parsing tasks with different formal programs (e.g., logic forms) and is also tested on *NON* text-to-SQL tasks (weakly-supervised WikiSQL and WikiTableQuestions where no programs are used, training is supervised by only answers/cell values).
> You can use any formal programs such as the logic form to design the SCFG and synthesize data with the corresponding programs labeled, and then use the data to pre-train GraPPa. We choose SQL as the formal program to represent the formal representation of the questions simply because more semantic parsing datasets are labeled in SQL.
> Also, we show the effectiveness of GraPPa on two even more challenging weakly-supervised semantic parsing tasks where training is supervised by answers (NOT any formal programs) in Table 5 for WikiTableQuestions (SOTA +1.8%) and Table 6 for weakly-sup. WikiSQL (SOTA +2.4%), in section 5-Analysis-Generalization, we discussed that despite that fact that GraPPa is pre-trained on SQL synthesized data, GraPPa is still able to improve performance on the two weakly-supervised semantic parsing tasks where the underlying distribution of programs (not necessarily in the form of SQL) are latent.
>
> **Are improvements of pre-training methods marginal?**
> As Reviewer 1, 2, and 4 mentioned, GraPPa consistently improves the performance over four popular/competitive (e.g, already >20 models reported for Spider) and representative (fully- and weakly-supervised) semantic parsing tasks in a significant margin. Results in Table 3-6 show that GraPPa can still improve the performance of SOTA base systems on these competitive tasks.
> 1. GraPPa outperforms RoBERTa-Large by +3.7% on Spider, 3, +1.8% on WikiTableQuestions, and +2.4% on weakly-sup. WikiSQL.
> 2. Moreover, on the low-resource setting, GraPPa outperforms RoBERTa-Large by 3.0% in fully-sup. WikiSQL and 3.9% in WikiTableQuestions.
>
> WikiTableQuestions and weakly-sup. WikiSQL tasks have no SQL label. They are weakly-supervised tasks where programs are latent and training is supervised only by answer results.

---

### Official Review · AnonReviewer2 · 2020-10-29

**Rating:** 7
**Confidence:** 4

**Review:**

## After author responses
Based on the revision of the draft and the authors' responses to the review, I am raising my score to 7 from 6.

---
## Overall summary
The paper proposes a pre-training method useful for training neural semantic parsing models that translate natural language questions into database queries (text-to-SQL). The authors manually write down and then sample from synchronous context-free grammars that can generate SQL queries along with corresponding natural language (but stilted) utterances that have the same meaning. The authors also collect questions about databases and tables from various sources. The combination of the two is used for fine-tuning RoBERTa using an auxiliary objective, then using the fine-tuned RoBERTa in semantic parsing tasks. The authors show strong results on four benchmarks.

## Strengths
- The authors obtain state-of-the-art results on four different benchmarks which convincingly shows that the proposed methods can help with empirical results.
- The method is simple and straightforward to implement.
- The paper addresses a problem domain with significant practical applications and community interest.

## Weaknesses
- It is unclear how much effort is needed to construct the SCFG. Since the SCFG was constructed directly by examining examples in Spider, the kinds of natural language questions and queries it generates may also be unfairly biased towards the distribution used in Spider.
- There could have been more quantitative analyses of the method and its component parts in the paper.

## Recommendation
I am giving a rating of 6 considering the strong results but the relative lack of analysis of the method (ablations, etc). I think further revisions of the paper would benefit from greater analysis of the method.

## Questions
- What would happen if you use the synthetic Spider data and use it to train the semantic parsing model for Spider?
- Does the grammar provide for multiple natural language utterances that will translate to the same SQL?
- The introduction states that "Our approach dramatically reduces the training time and GPU cost." Is there evidence to show this beyond Figure 2? There, the gains don't seem so dramatic. Considering that fine-tuning GRAPPA took 10 hours on 8 V100 GPUs, it seems unlikely that there will GPU cost will be reduced especially if the fine-tuning cost is also included.
- What happens if the task-specific training data is also used with the MLM or SSP objectives in pre-training?  https://arxiv.org/abs/2004.10964 gives evidence that it can be useful to fine-tune RoBERTa on the downstream task's data using the pre-training objectives.

## Miscellaneous comments
The papers for Tapas, Overnight, and TaBERT are duplicated in the references.

---

> ### Author Response · Authors · 2020-11-16
> **Thank you for your review**
>
> Thank you for your detailed comments!
>
> **W2: Does the paper lack quantitative analyses of the method (ablations, etc)?**
> As reviewer 1 and 4 mentioned, we do provide an ablation study of our proposed pre-training objectives (MLM vs. SSP vs. MLM+SSP) on all four semantic parsing tasks (Table 3 for Spider, 4 for fully-sup WikiSQL, 5 for WikiTableQuestions, and 6 for weakly-sup WikiSQL). Below is a short summary of the ablation study in our paper:
> 1. GraPPa (pre-trained with our MLM+SSP) vs. RoBERTa-Large: GraPPa pre-trained with MLM+SSP significantly outperforms RoBERTa-Large on all four tasks. E.g., +3.7% on Spider, +1.8% on WikiTableQuestions, and +2.4% on weakly-sup. WikiSQL.
> 2. MLM+SSP vs. (MLM or SSP): GraPPa pre-trained with MLM+SSP objectives consistently outperforms the one trained with one of them. E.g., ~+2% on Spider (Table 3), ~1.5% on WikiTableQuestions (Table 5), and ~+1.2% on weakly-sup WikiSQL (Table 6).
> 3. GraPPa pre-trained with MLM or SSP only vs. RoBERTa-Large: MLM and SSP only also consistently outperform RoBERTa-Large on all the four tasks based on results in Table 3-6.
>
> Also, in section 5-Pre-training-time-and-data, we show that the best performance of GraPPa is usually achieved by pre-training with MLM+SSP for only 5 epochs on our relatively small synthesized data and natural questions over DB. We also plan to vary the ratio of MLM and SSP data in our pre-training set to observe the change in model behavior. However, given the time and energy consumption of this experiment, we decided to leave it out of the scope of this work. Our main contribution is to show that synthetic data can be effectively leveraged for table semantic parsing pre-training when regularized with standard MLMs.
>
> **W1: how much effort is needed to construct the SCFG?**
> We didn’t specifically count the time. It might take only a few hours (3 - 5 hours) for the manual step in constructing the SCFG presented in this work. We wrote a script to automatically group and count Spider training examples by program templates, and then select the top ~90 templates. For the selected templates, we then manually create the alignments between some questions (usually ~3-10) and program templates, which approximately takes a few hours. About ~500 Spider examples are studied to induce the SCFG.
>
> **W1: SCFG is unfairly biased towards the distribution used in Spider.**
> We design our SCFG only on Spider on purpose (Section 5-Analysis-Generalization) because we want to show that despite the fact that GraPPa pre-trained on such corpus is optimized to the Spider data distribution, which is very different from WikiSQL and WikiTableQuestions (weakly-sup task), GraPPa is still able to improve performance on the two tasks. The reason why we choose Spider to induce the grammar is that Spider covers more production rules than others.
>
> **Q1: what if training a Spider model on the synthetic Spider data?**
> Recent work (Zhang et al., 2019b; Herzig et al., 2020b; Campagna et al., 2020; Zhong et al., 2020) in semantic parsing show that training models on a combination of the extra synthetic data and original training data does NOT improve or even HURT the performance (Section 5-Pre-training vs. training data augmentation). We also find that combining Spider training data and ~10k synthetic data doesn’t significantly improve the performance.
>
> **Q2: multiple natural language utterances for the same SQL?**
> Yes, we have about 3-10 questions for each SQL template in our SCFG.
>
> **Q3: Does the proposed pre-training reduce the training time and GPU cost**
> Sorry about the possible confusion here. We actually compare our GraPPa pre-training strategy with other recent pre-training methods for semantic parsing such as TaBERT and TAPAS. *To SOTA the four tasks, GraPPa uses <10 hours on only 8 16GB Nvidia V100 GPUs vs. 6 days on >100 V100 GPUs/3 days on 32 TPUs for TaBERT and TAPAS*.
>
> **Q4: what happens if the task-specific training data is also used with the MLM or SSP objectives in pre-training?**
> Although we did not do the same experiments, we would like to point to the [RAT-SQL paper](https://arxiv.org/pdf/1911.04942.pdf) for some suggestions. They added a similar alignment loss (similar to SSP) on the Spider training data and found that it didn’t make a statistically significant difference (in Appendix B).

---

### Official Review · AnonReviewer1 · 2020-10-30
**Nice idea of using synthetic data to improve compositional textual understanding**

**Rating:** 7
**Confidence:** 4

**Review:**

This paper presents a general-purpose pre-training approach for jointly encoding utterances and relational tables in the task of table semantic parsing, where a natural language utterance is transduced into an executable query (e.g., SQL) over relational database tables. A core challenge in table semantic parsing is to understand the compositional semantics in utterances, and further ground salient entities and relations in the utterance onto the corresponding tabular schema (e.g., columns, cells). To improve understanding and grounding of compositional utterances, the authors propose fine-tuning a pre-trained masked language model (RoBERTa) using linearized table headers paired with synthetic utterances generated from a synchronous context free grammar, via an objective that encourages the model to discover the syntactic roles of columns mentioned in the input utterance. Experiments over four datasets demonstrated strong results when the this newly proposed pre-training objective is combined with classical masked language modeling objective.

Strong Points:

* The paper is in general well written and easy to follow;
* The idea of fine-tuning ``monster'' pre-trained language models over massive open-domain textural corpora using domain-specific synthetic corpora and objectives could also inspire future research on applying pre-trained LMs in low-resource domains with heterogenous (semi-)structured data.
* Results over four semantic parsing benchmarks provide convincing story of the effectiveness of the proposed approach.


Detailed Comments:

[Pre-training Logical Form Coverage] The production rules ($\beta$'s) in Table 1 are high-level SQL sketches with fixed level of compositionality. While this would help improve the quality of canonical utterances, this will would limit the expressiveness of the grammar, since those production rules cannot be composed or chained. I was wondering if the authors had considered using a more compositional grammar (e.g., the general grammars in Wang et al. 2014 or Herzig and Berant, 2020) with larger logical form coverage.

Additionally, the 90 productions used in generating synthetic data are curated from utterances in the Spider dataset. While it is not the focus of the paper, incorporating rules learned from a more broad collection of datasets would be helpful, as suggested by the experimental results, where RoBERTa tuned with the proposed SQL semantic prediction objective (GRAPPA+SSP) significantly outperforms training with the classical masked language modeling objective (GRAPPA+MLM) on Spider (Table 3), whose SQL program templates are covered by the synthetic data during pretraining, while GRAPPA+SSP performs on par (Table 6) or slightly underperforms GRAPPA+MLM (Tables 4, 5) on other benchmarks. This is not surprising, as programs in those datasets have significantly different underlying patterns compared to Spider's. For example, WikiTableQuestions has more `argmax` queries, and queries sensitive to the position of rows (e.g., "What is the host city of the first Olympics?"). This might suggest increasing logical form coverage is important for further improving the performance.

[Interplay between the SSP and MLM objectives] In section 1, the authors mentioned the importance of carefully balancing between preserving the original natural language representations and injecting the compositional inductive bias through our synthetic data, without details of the pre-training procedure. Are these two objectives used in a particular order, or are MLM and SSP examples sampled randomly during pre-training?

The experimental results suggest that the key is to combine SSP+MLM, it would be interesting to study the relative impact of the two objectives, e.g., by varying the amount of their respective pre-training examples.

[Encoding Table Contents] The proposed approach only encodes table header as input to the Transformer, while ignoring its contents. Previous works have demonstrated the importance of encoding table contents relevant to the input utterances in semantic parsing. While the downstream parser might have features to capture content information (e.g., a binary feature that indicates if a cell in a column is mentioned in the utterance), it would be useful to directly encode such information during pre-training.

---

> ### Author Response · Authors · 2020-11-16
> **Thank you for your review**
>
> Thanks for your detailed and thoughtful comments!
>
> **Pre-training Logical Form Coverage**
> Yes, we agree that a more compositional grammar would cover more logic forms, and potentially benefit the LMs pre-trained on its induced synthetic data. However, we also have a doubt that compositional grammar might also generate a lot of logic forms that would not usually appear in reality, which then bias pre-train LMs towards an unrealistic distribution. A careful algorithm design might resolve this problem. But there are no corresponding natural questions for complicated corner program cases to build the SCFG (or it is usually hard to annotate such questions).
> Instead, the main focus of our paper is to try a relatively simple yet effective solution to show the effectiveness of leveraging synthetic data in semantic parsing pre-training: we wrote a script to automatically group and count Spider training examples by program templates, and then select the top ~90 templates. For the selected templates, we then manually create the alignments between some questions (usually ~3-10) and program templates, which approximately takes a few hours. About ~500 Spider examples are studied to induce the SCFG.
>
> **Incorporating rules learned from a more broad collection of datasets**
> Grammar induced by a small set of Spider data actually covers the logic forms in WikiSQL and the majority of those in Spider and WikiTableQuestions (~80%). We agree that incorporating rules from a broader range of datasets (e.g. WikiTableQuestions) would further improve the performance. However, as discussed in our generalization analysis (section 5), we study rules from only the Spider dataset on purpose to show the generality of our method. Despite being trained on a distribution close to the Spider dataset, GraPPa is able to improve performance on WikiSQL and WikiTableQuestions, both have very different distributions compared to Spider.
>
> **GraPPa+SSP doesn’t outperform GRAPPA+MLM that much on WikiSQL and WikiTableQuestions**
> These results actually contribute to our key argument in the paper: *In order to effectively inject compositional inductive bias to LM, pre-training on synthetic data should be regularized properly  (using SSP+MLM together instead of SSP or MLM only) in order to balance between preserving the original BERT encoding ability and injecting compositional inductive bias, otherwise, the improvements are not robust and limited (using SSP or MLM only)*. More specifically, GraPPa pre-trained with MLM+SSP consistently and significantly outperforms the one pre-trained with MLM or SSP separately (MLM+SSP vs. MLM only or SSP only: E.g., about +2% on Spider, +1.5% on WikiTableQuestions, and +1.2% on weakly-sup WikiSQL.).
>
> **Interplay between the SSP and MLM objectives**
> Thanks for pointing this out! We will update the rebuttal version with more detail on sample scheduling in the pre-training procedure. Basically, we mix natural language examples and synthetic examples together as the final pre-training data. The examples in these two groups are randomly sampled and MLM loss is computed if the selected example is a natural language question, otherwise SSP for a synthetic example.
> We also believe that it would be interesting to study the relative impact of the two objectives by varying the respective number of pre-training examples. In the current paper, we compare the results of two extreme cases and the mixture (0 MLM+1SSP, 1 MLM + 0 SSP, and roughly 1/2 MLM + 1/2 SSP), using 391k examples and 475k SSP examples in total. In future work, we plan to conduct more studies by varying the ratio in the mixture.
>
> **Encoding Table Contents**
> We do use table content when synthesizing our pre-training data. For example, if the SQL template contains a value (e.g., VALUE0 in COLUMN0 = VALUE0), we will sample a value for the selected column from the table content and add it to the SQL and question templates. This way during pre-training, GraPPa can access some table contents and learn the linking between values and columns. Just as you mentioned, maybe an objective designed more specifically for modeling table content would help more.
> Also, directly encoding the entire rows of table contents might hurt the language model pre-training since the table content usually contains a lot of numbers and noisy entries. Careful filtering and cleaning have to be implemented.

---

### Author Response · Authors · 2020-11-25
**Summary response to all reviewers and the new revision**

We greatly thank all the reviewers for their feedback and constructive comments. We are pleased that the reviewers appreciate the simplicity/effectiveness of our method (R1, 2), find our experiments to be solid/extensive/convincing (R1, 2, 3, 4), and think our idea is inspiring for future research (R1). We’ve revised and updated the draft which reflects the reviewers' comments. The updates are summarized as follows:
- Section 2.2: more detailed SCFG construction in data synthesis. (R1,2)
- Section 2.4: add more details on the pre-training procedure. (R1)
- Section 5 (pre-training objectives) and Section 4 (Overall results): Emphasis more on the effectiveness of MLM+SSP. (R2)
- Section 5 (pre-training objectives): add pre-training time comparison with TaBERT and TAPAS. (R2)
- Section 5 (generation): add a discussion on why our method is NOT designed for and only tested on text-to-SQL tasks. (R3)
- Section 7: more discussions on potential interesting future work. (R1)
- Appendix A.1: add a discussion on adding SSP/MLM objective in task-specific training. (R2)

---

### Decision · Program_Chairs · 2021-01-07
**Final Decision**

**Decision:**

Accept (Poster)

**Comment:**

This paper proposes a pre-training technique for semantic parsing with an emphasis on semantic parsing and the technical details required to actually make it work in practice. Overall, all reviewers agree that the results are very good, and you see nice improvement across multiple text-to-SQL datasets. Some reservations have been raised on (a) the difficulty of creating the SCFG for generating the synthetic data, but this seems to have been properly addressed by the authors and requires a reasonable amount of effort. and (b) how tailored the pre-training task is to a particular task (text-to-SQL) and dataset (Spider). Overall, I tend to agree that the fact that one sees improvement on Spider is slightly less compelling as the grammar is derived from it, but the authors rightfully claim that consistent improvements are also evident in other datasets, even if the gains are somewhat smaller. One can hope the idea can also be generalized to other setups where synthetic data can be generated and the details of how to combine synthetic data with real data should be useful.